# Harvest Season and Morphological Variation of Canistel (*Pouteria campechiana*) Fruit and Leaves Collected in Different Zones of Mexico

Karen M. Granados-Vega [1], Silvia Evangelista-Lozano [2,*], Sandra L. Escobar-Arellano [2], Tomás Rodríguez-García [3], José F. Pérez-Bárcena [4] and Juan G. Cruz-Castillo [5]

1 Doctorado en Ciencias en Desarrollo de Productos Bióticos, Instituto Politécnico Nacional, Centro de Desarrollo de Productos Bióticos CEPROBI, Yautepec 62739, Mexico; kgranadosv1801@alumno.ipn.mx

2 Instituto Politécnico Nacional, Centro de Desarrollo de Productos Bióticos CEPROBI, Yautepec 62739, Mexico

3 Estancia Posdoctoral CONAHCyT, Instituto Politécnico Nacional, Centro de Desarrollo de Productos Bióticos CEPROBI, Yautepec 62739, Mexico; trodriguezg1300@alumno.ipn.mx

4 Instituto Politécnico Nacional, Centro Interdisciplinario de Ciencias de la Salud, Unidad Milpa Alta, Mexico City 12000, Mexico; jperezba@ipn.mx

5 Centro Regional Universitario Oriente, Universidad Autónoma Chapingo, Km 6.5, Carretera Huatusco-Xalapa, Huatusco, Veracruz 94100, Mexico; jcruzc@chapingo.mx

* Correspondence: sevangel@ipn.mx

**Abstract:** Canistel *(Pouteria campechiana* (Kunth) Baehni; yellow sapote, canistel) is an edible fruit that is native to Mexico; the tree is used as an ornamental and medicinal plant. Descriptive studies were carried out with the objective of characterizing the morphology of the fruit and leaves of trees located in different regions of Mexico to select outstanding specimens for large scale reproduction. The selection of trees was carried out in three geographically neighboring areas, where visible differences have been observed in fruit morphology: size, shape, color, number of seeds, and leaf shape. Semi-ripe fruit were collected from each area, weight and measurements were noted, seeds were extracted, and a data matrix was obtained: (fruit weight (grams), fruit polar diameter (centimeters), fruit equatorial diameter (centimeters), pulp weight (grams), number of seeds, seed weight (grams), seed polar diameter (centimeters), seed equatorial diameter (centimeters), and total soluble solids (SST)). Leaves were collected at the mature stage, and an image analysis was carried out to evaluate morphometric parameters: area, perimeter, major diameter, minor diameter, circularity, solidity, and angle. An analysis of variance was performed to find significant differences in the fruit data and a principal component analysis was performed with the leaf data. In Zone 2, the fruit had three seeds, the highest weight in all the zones sampled (180 to 330 g), and the greatest amount of pulp (198.88 g); they were subglobose in shape, and 33% soluble solids. There were eight months to harvest in Zone 1 and nine months in Zone 3. The main variables in the principal component analysis were leaf area, perimeter, and minor diameter; 84.6% of the variability represented by the first three components is sufficient to explain the difference between the leaves of the three zones. This data can be used to assess the propagation of canistel in areas with favorable climatic conditions for fruiting in tropical and subtropical regions.

**Keywords:** yellow sapote; tree selection; viviparity; leaf characterization; early seed germination

## 1. Introduction

With the effects of climate change, it has become important to focus research on obtaining seeds with high productive potential. National and international markets now demand species with high nutritional and medicinal potential. In Mexico, one important plant species is *Pouteria campechiana* (Kunth) Baehni (synonymy: *Lucuma salicifolia* Kunth, *Lucuma campechiana* Kunth, *Lucuma nervosa* A. DC, *Achras lucuma*), commonly known as egg fruit, canistel, drunken sapote, or yellow sapote [1]. These kinds of trees are classified by

Colunga-García-Marín and Zizumbo-Villarreal as non-domesticated fruit trees [2]. These trees are mainly propagated by seed, but it is also possible to propagate them asexually with cuttings [3]. Mexico and Central America are considered origin centers of this species [4]. According to Gonzalez (2004) [5], the family Sapotaceae has been extensively recognized since pre-Columbian times, with depictions of the fruit found on clothing and clay pots.

Analysis of canistel has found polyphenols with antioxidant activity in the canistel fruit [6] and flavonoids (myricetin, quercetin), phenolic compounds, terpenes, and other compounds in seeds and leaves. Ethanolic extracts of canistel seeds and leaves are used in alternative medicine; phenolic compounds, such as gallate acid, have anti-inflammatory properties and are useful in treating gastrointestinal disorders [7]. Additionally, they possess antioxidant properties due to the presence of flavonoids, such as myricetin and quercetin [8,9]. These bioactive compounds and their concentrations may differ, as with the differences in phenols reported by Ma et al. [10] and Kong et al. [11]; these differences may be attributed to environment, season, soil, or management [12].

There are no canistel plantations in Mexico; the trees are cultivated in domestic gardens, in both urban and rural areas. The number of trees has decreased with population growth and construction, and the species is threatened with extinction.

These references support the examination of canistel in Mexico, where specimens still exist in backyard cultivation in the state of Morelos. There are specimens in domestic gardens in the municipalities of Yautepec and Jiutepec, Morelos. There are also 30 trees in the Emiliano Zapata Experimental field, on the south side of the municipality of Yautepec, which are branch one of seeds from Jiutepec [13]. The ornamental, medicinal, and nutritional qualities of canistel make it important to describe these trees with an eye toward implementing their propagation by seed, maintaining diversity, and finding the trees that are best adapted to overcoming the adversities of climate change. To propagate these trees effectively and maintain diversity, it is crucial to provide detailed descriptions of their characteristics in local areas. Additionally, considering climate change, identifying the best adapted trees in these regions is essential for their survival. The information obtained will support the planting of this tree in other areas with similar climates, and it can be propagated in public spaces, gardens, educational centers, and in association with other fruit trees.

The objective of this study was thus to characterize the fruit, seeds, leaves, and harvest time of canistel trees located in these areas of Mexico.

## 2. Materials and Methods

### 2.1. Study Zones

Canistel collection regions were geolocated (Table 1), and the climate, temperature, and rainy-day data for the different areas were obtained for the average of each month over a period of 20 years, using the Köppen–Geiger classification (Figure 1). The study locations are situated in Yautepec (Zone 1 and Zone 3) and Jiutepec (Zone 2), each exhibiting distinct climatic features. The study took place between 2018 and 2022 in Zones 1 and 2, and between 2022 and 2023 in Zone 3. Zone 3 (2022–2023) corresponds to the first subdivision of seeds from Zone 2. One tree was selected from each area, specifically those in garden cultivation, due to their scarcity compared to those in backyard cultivation.

The difference in precipitation between the months with the lowest and highest levels is 267 mm. Average temperatures vary throughout the year by 5.8 °C. The month with the highest relative humidity is September (79.21%), and the month with the lowest is March (37.46%). The month with the greatest number of rainy days is August (26.30), and the month with the lowest number is December (1.57).

In Zones 1 and 3 (Figure 1A), there is a greater amount of rainfall, whilst in Zone 2 (Figure 1B) the rainfall is lower and varies throughout the period from April to October.

**Table 1.** Climatic and geographic features of the three collection areas of canistel (*Pouteria campechiana*) fruit and leaves considered as a phylogenetic resource in Mexico.

| Descriptive Characteristics | Study Zones | | |
|---|---|---|---|
| | Zone 1 | Zone 2 | Zone 3 |
| Geo location | 18.87595 LN-99.077464 LW | 18.884546 LN-99.17745 LW | 18.824988 LN-99.096042 LW |
| Locality | Yautepec de Zaragoza | Jiutepec | Yautepec de Zaragoza |
| Neighborhood | Otilio Montaño | Jiutepec Downtown | San Isidro |
| m.a.s.l. [1] | 1210 | 1350 | 1059 |
| Weather | Semi-/sub-humid (66% of the year), the warmest of the temperate climates. Warm sub-humid (34%), with summer rains, the driest of the sub-humid climates, little fluctuation. | Semi-warm and sub-humid (28%), the coolest of the warm climates. Warm sub-humid with summer rainfall (72%), the greater part of the year. | Warm sub-humid (100%), dry (low deciduous forest). |
| Precipitation | Medium sub-humid in summer, with rainfall concentrated in that season. Dry from November to April. | Average annual rainfall is 1021 mm; 890 mm from June to October, the rainy season. | Summer rainfall; winter rainfall less than 5%. |
| Temperature | Hot summer (34 ± 4 °C), extreme nighttime temperature drops (18 ± 3 °C); Ganges-type temperature regime (highest temperature in May, before summer solstice and rainy season. | Isothermal (the average thermal variation of the year does not exceed 3 °C), Ganges-type temperature regime, ranging from 11 °C to 32 °C, with an average of 21.2 °C a maximum average variation of 31.4 °C, and an absolute maximum of 39.8 °C. Hottest months are April and May; coolest months are December and January. | Extreme hot summer days (35 ± 3 °C); evening temperature drops (20 ± 3 °C); rest of the year cooler but with similar drop in evening. |
| Harvest season (Months) | November–December | May–July | March–April |
| Early seed germination (%) | 0.2 | 30 | 0.1 |

[1] m.a.s.l.: meters above sea level [14].

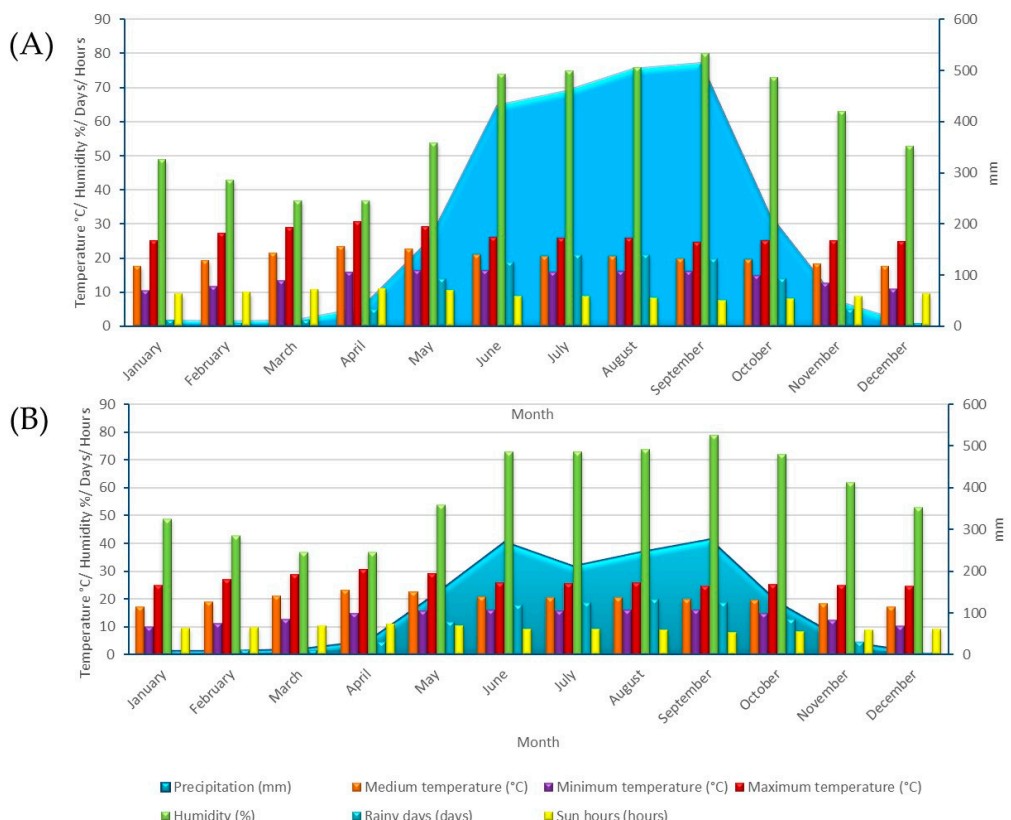

**Figure 1.** Historical climate data in the sample zones (temperature, humidity, precipitation) in the collection zones of canistel (*Pouteria campechiana*): zone 1 and 3 (**A**) and zone 2 (**B**).

## 2.2. Plant Material: Growing Conditions and Selection

After identifying the trees to be sampled (one tree per zone, Zone 1 containing a 25-year-old tree, Zone 2 containing a 35-year-old tree, and Zone 3 containing a 10-year-old tree) through continuous walks and monitoring during fruiting, the fruit and leaves were collected. Once the trees to be sampled were chosen, their fruit and leaves were harvested. The maturity criterion for fruit harvest was semi-ripe [15]. The leaves collected were mature and homogeneous in color without phytosanitary problems; both leaves and fruit were transported in a cooler to the laboratory and placed on a table to cool [16]. The fruit and leaves were cleaned with a damp cloth. Once dry, the leaves were placed between thin transparent glass plates for 6 h so that they would not lose their shape.

The fruit and leaf collection were monitored from anthesis and took a total of 11 months in Zone 2, 8 months in Zone 1, and 9 months in Zone 3.

## 2.3. Characterization of Fruit

Canistel fruit harvested were numbered and weighed on a digital balance (MediaDeta DS-5). The polar diameter (from the base to the distal part or protuberance of the fruit) and the equatorial diameter (at the widest part of the fruit) were measured with a digital caliper (Titan model 55674). Then, both pulp and seeds were weighed, and the number of seeds with premature germination was recorded. Total soluble solids (TSS), which indicate sucrose content and includes carbohydrates, organic acids, proteins, fats, and minerals of the fruit, were measured with a refractometer (Fisher) [17].

## 2.4. Characterization of Leaf Morphology

Images of the adaxial and abaxial faces of five mature leaves (Figure 2A) were captured in RGB using a digital camera (FUJIFILM FinePIX S8600, f = 4.5–162 1:2 9–6.9 lens, Japan) connected to a computer and stored in JPG format at 1322 × 3128 pixels. These images

were converted to a grayscale format using ImageJ version 1.5.3 software to set the correct scale (Analyze > Set Scale > adjust the actual distance) and the threshold tool (from $5 \pm 1$ to 79) was used to segment the image (Image > Threshold > Apply). The dimensions of the segmented image were measured in mm, using the Analyze > Measure tools. The results were arranged in a data matrix for processing and analysis. The following variables are indicated in Figure 2:

- Area: the number of pixels within the shape bounded by the perimeter.
- Perimeter: the number of pixels forming the boundary of the leaf.
- Major diameter: length of the leaf.
- Minor diameter: width of the leaf.
- Circularity: the ratio $4\pi A/p2$, excluding local irregularities (equal to 1 for a circular object and less than 1 for non-circular object).
- Solidity: pixel density, or the number of pixels joined without gaps; an object with greater solidity has fewer gaps, and an object with less solidity has more gaps.
- Leaf angle: evaluated by manually segmenting and marking the main and secondary veins of each leaf on the abaxial face, using Freehand line > Draw > Measure.

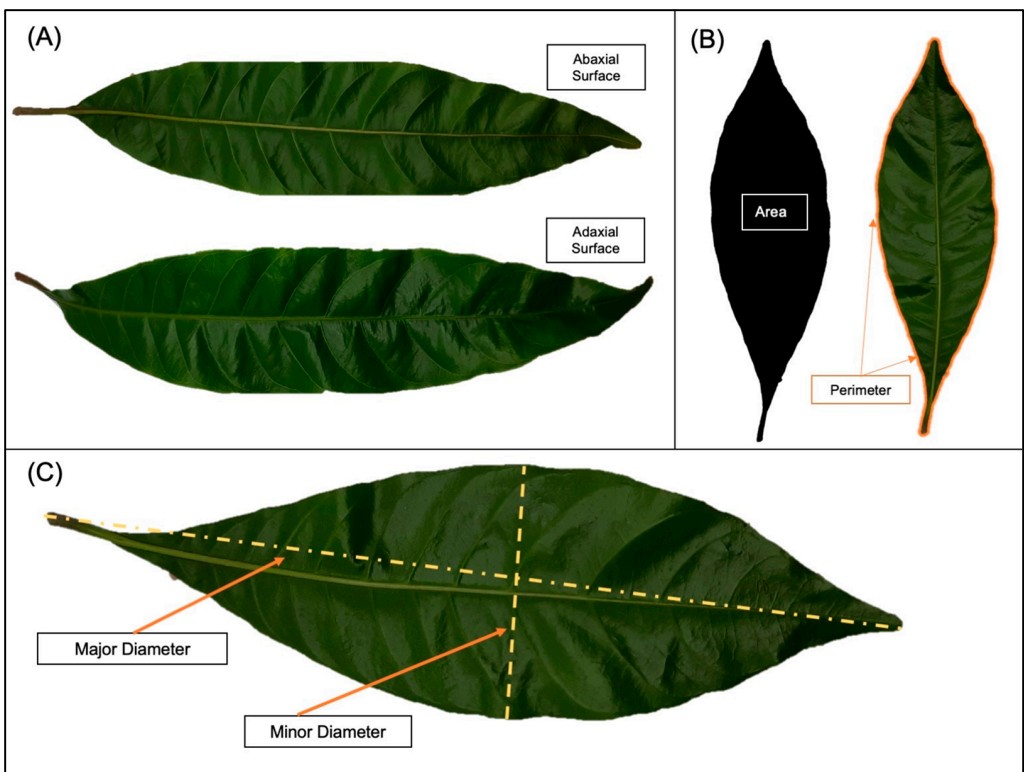

**Figure 2.** Canistel tree leaf with description of the evaluated variables. (**A**) front and back side of the leaf, (**B**) size variables and (**C**) length and width variables.

### 2.5. Statistical Analysis

Data from the climate, fruit, and leaf characteristics were analyzed with a one-factor analysis of variance (ANOVA) with $p \leq 0.05$ and a Bonferroni multiple comparisons test [18]. The means and standard errors were calculated using SPSS version 20. The data matrix of the leaves was standardized, and a multivariate analysis was performed to determine the principal components using Minitab version 17.

## 3. Results

### 3.1. Study Zones

The features of the three canistel leaf and fruit collection regions were examined. Climatic characteristics for each study zone are presented in Table 1 and Figure 1. This

information is utilized to comprehend morphological differences in the leaves and fruit. Zone 1 is situated at an altitude of 1210 m above sea level, Zone 2 at an altitude of 1350 m above sea level, and Zone 3 at an altitude of 1050 m above sea level. Zones 1 and 3 experience higher and more consistent precipitation from June to September and have higher temperatures than Zone 2. In contrast, Zone 2 has less precipitation with less consistency from May to October.

### 3.2. Characterization of Fruits

The fruit of Zone 2 presented the greatest weight (180–330 g) and quantity of pulp (198.88 g), with a sub-globose shape. They contained approximately three seeds with an average weight of 19.71 g. They showed statistically significant differences in polar and equatorial diameter and total soluble solids from the fruit in Zones 1 and 3 (Table 2).

The time from anthesis to harvest was eleven months in Zone 2, eight months in Zone 1, and nine months in Zone 3; the morphological variation of the fruit in the three zones can be seen in Figure 3.

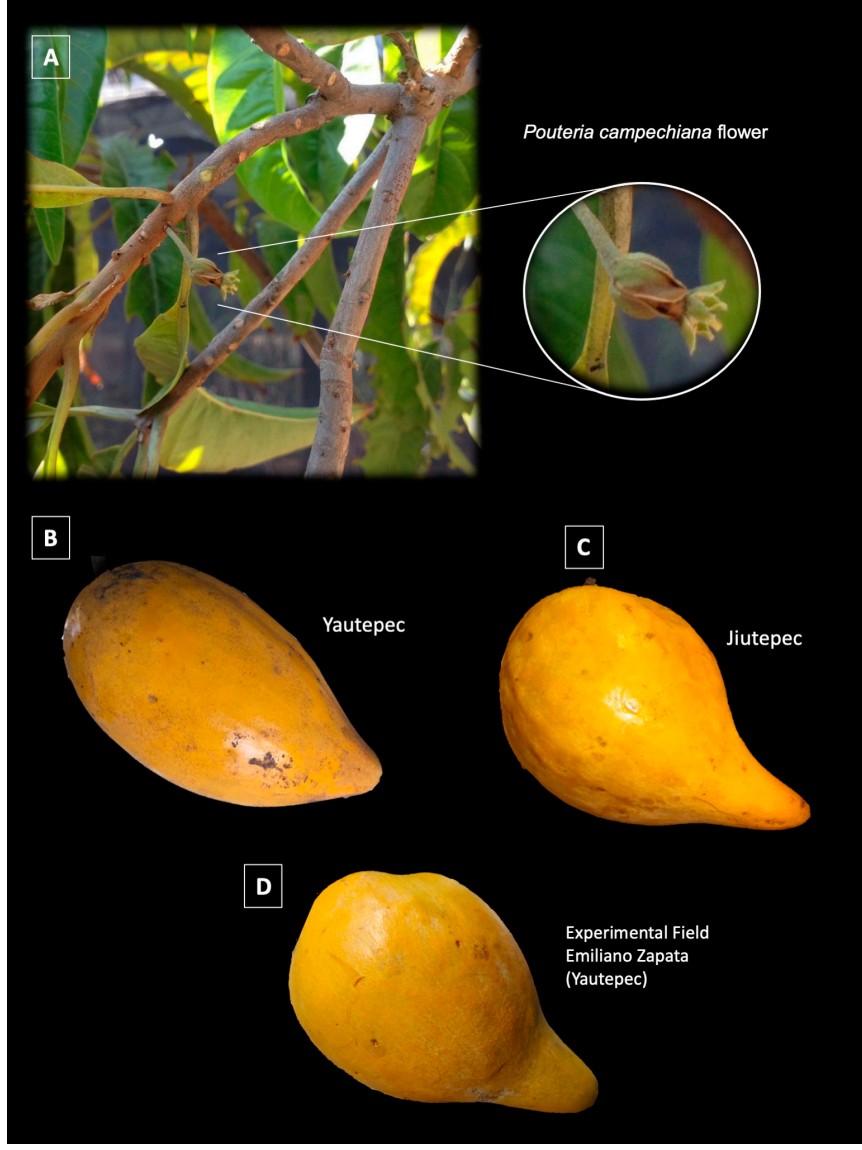

**Figure 3.** (**A**) Canistel (*Pouteria campechiana*) flower at anthesis of and fruit of the three zones, (**B**) Zone 1, (**C**) Zone 2 and (**D**) Zone 3.

**Table 2.** Morphological and chemical features of canistel (*Pouteria campechiana*) fruit obtained from three locations in Mexico.

| Parameters | Zone 1 Mean | Zone 2 Mean | Zone 3 Mean |
|---|---|---|---|
| Fruit weight (g) | 146.24 ± 25.11 [a] | 261.94 ± 76.78 [b] | 160.18 ± 51.64 [a] |
| Fruit polar diameter (cm) | 9.49 ± 1.16 [a] | 11.03 ± 0.84 [b] | 9.07 ± 0.97 [a] |
| Fruit equatorial diameter (cm) | 5.49 ± 0.46 [a] | 7.29 ± 0.96 [b] | 6.03 ± 0.93 [a] |
| Pulp weight (g) | 122.76 ± 27.62 [a] | 198.88 ± 54.39 [b] | 101.26 ± 35.87 [a] |
| Number of seeds | 1.12 ± 0.33 [a] | 2.59 ± 0.87 [b] | 1.29 ± 0.47 [a] |
| Seed weight (g) | 10.12 ± 2.26 [a] | 19.71 ± 5.68 [b] | 19.78 ± 3.29 [b] |
| Seed polar diameter (cm) | 4.29 ± 0.45 [b] | 4.49 ± 0.26 [b] | 5.50 ± 0.57 [a] |
| Seed equatorial diameter (cm) | 2.08 ± 0.28 [b] | 2.24 ± 0.17 [a,b] | 2.40 ± 0.22 [a] |
| Total soluble solids (SST) | 27.47 ± 2.79 [a] | 33.18 ± 0.39 [b] | 26.12 ± 1.22 [a] |

Note: Values in the same row that do not share the same superscript letter are significantly different at $p < 0.05$ in the test for bilateral equality of column means. The tests assume equal variances, using the Bonferroni correction.

### 3.3. Characterization of Leaf Morphology

Differences in leaf area were statistically significant for the three zones; Zone 3 had the largest leaves. Leaves from Zone 3 had a larger perimeter than those from Zones 1 and 2. The major diameters (length) were equal for fruit from Zones 1 and 3 and significantly different from those in Zone 2; the minor diameter in the three zones was significantly different. The circularity index for the three zones was different; the soundness index and angle variable were different in Zone 3 than in Zones 1 and 2 (Table 3).

**Table 3.** Morphological characteristics of leaves collected in three zones of Mexico.

| Parameters | LEAVES | | |
|---|---|---|---|
| | Zone 1 | Zone 2 | Zone 3 |
| | Mean | Mean | Mean |
| Area | 52.52 [b] | 48.95 [a] | 63.54 [c] |
| Perimeter | 44.79 [b] | 46.73 [a] | 48.46 [c] |
| Major diameter | 16.14 [b] | 16.52 [a] | 15.99 [b] |
| Minor diameter | 4.10 [b] | 3.77 [a] | 5.07 [c] |
| Circularity | 0.33 [b] | 0.28 [a] | 0.34 [c] |
| Solidity | 0.92 [a] | 0.92 [a] | 0.90 [b] |
| Angle | 48.46 [a] | 45.42 [a] | 56.67 [b] |

Note: Values in the same row that do not share the same subscript letter are significantly different at $p < 0.05$ in the test for bilateral equality of column means. Tests assume equal variances, using Bonferroni correction.

### 3.4. Correlation Analysis

Analysis of the morphological characteristics of the canistel leaves showed a very high correlation between area and minor diameter, a high correlation between perimeter and major diameter, and between minor diameter and circularity. There was a moderate correlation between area and circularity, and also between area and perimeter (Table 4).

**Table 4.** Pearson correlation of the variables measured in the canistel leaves of the three study zones.

|  | Area | Perimeter | Major Diameter | Minor Diameter | Circularity | Solidity |
|---|---|---|---|---|---|---|
| **Perimeter** | 0.643 0.000 ** | | | | | |
| **Major diameter** | 0.515 0.000 ** | 0.76 0.000 ** | | | | |
| **Minor diameter** | 0.911 0.000 ** | 0.399 0.000 ** | 0.12 0.017 * | | | |
| **Circularity** | 0.552 0.000 ** | −0.279 0.000 ** | −0.171 0.001 ** | 0.7 0.000 ** | | |
| **Solidity** | −0.291 0.000 ** | −0.466 0.000 ** | −0.076 0.129 NS | −0.324 0.000 ** | 0.162 0.001 ** | |
| **Angle** | 0.277 0.000 NS | 0.018 0.724 ** | −0.07 0.163 ** | 0.356 0.000 NS | 0.331 0.000 NS | −0.124 0.013 NS |

Not significant NS and significant at a $p \le 0.05$ * and $p \le 0.001$ **.

### 3.5. Principal Component Analysis

The eigenvalues of the principal component analysis indicated that the first principal component contributes 43.7% of the total variability, the second component 28.1%, and the third component 14.5%; the combination of these three components thus describes 86.4% of the variability of the data (Table 5).

**Table 5.** Eigenvalues, absolute, and cumulative percentage of variance of the principal components of the canistel leaves analysis.

| Principal Component | Eigenvalue | Variance (%) | Cumulative Variance (%) |
|---|---|---|---|
| 1 | 3.0615 | 43.7 | 43.7 |
| 2 | 1.9676 | 28.1 | 71.8 |
| 3 | 1.0175 | 14.5 | 86.4 |
| 4 | 0.7358 | 10.5 | 96.9 |
| 5 | 0.2118 | 3 | 99.9 |
| 6 | 0.0045 | 0.1 | 100 |
| 7 | 0.0013 | 0 | 100 |

With respect to the eigenvectors (Table 6), the variable with the greatest contribution to the first component was area (0.559), followed by minor diameter (0.507); to the second principal component were perimeter (0.476) and circularity (−0.588); and to the third component were the solidity (−0.765) and major diameter (0.434).

**Table 6.** Eigenvectors of the first three principal components of the canistel leaf analysis.

| Variable | Principal Components | | |
|---|---|---|---|
| | 1 | 2 | 3 |
| Area | **0.559** | −0.056 | −0.174 |
| Perimeter | 0.408 | **0.476** | 0.006 |
| Major Diameter | 0.300 | 0.454 | **−0.434** |
| Minor Diameter | **0.507** | −0.270 | 0.029 |
| Circularity | 0.256 | **−0.588** | −0.243 |
| Solidity | −0.254 | −0.216 | **−0.765** |
| Angle | 0.212 | −0.313 | 0.370 |

The two-dimensional diagram shows the projection of the variables on the first two principal components. The variables projected on the plane match the results obtained, where area and minor diameter show the greatest contribution to the variation in the first component, and perimeter and major diameter in the second component (Figure 4).

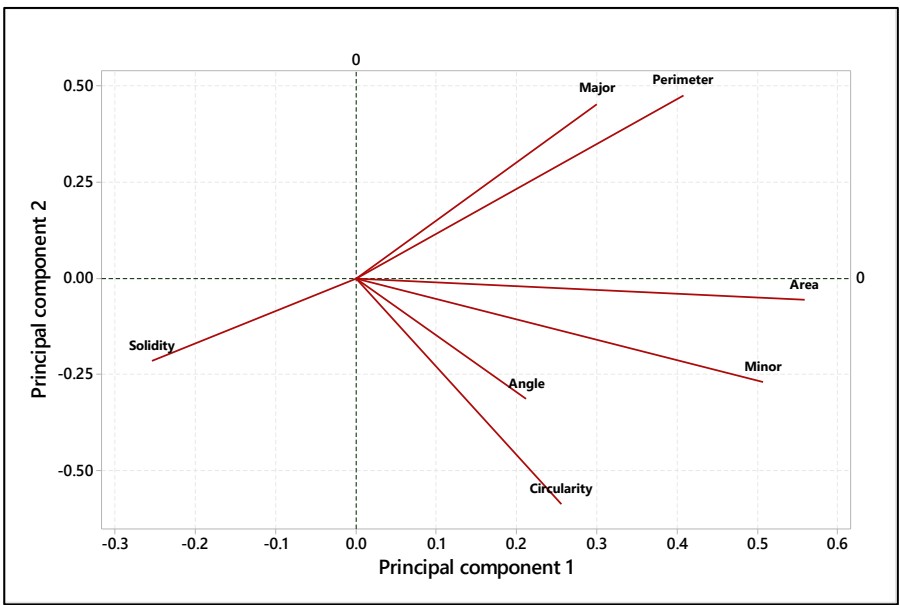

**Figure 4.** Two-dimensional diagram for the variables measured on the leaves, showing the projection of the zones on the first two principal components.

## 4. Discussion

The altitude above sea level and climate were different in the three study zones (Table 1). Zone 2 had the highest altitude (1350 m.a.s.l.), followed by Zone 1 (1210 m.a.s.l.) and 3 (1059 m.a.s.l.). Pennington and Sarukhán [19] report that canistel has been found at sea level and up to 1400 m.a.s.l. in tropical and subtropical climates, so all three zones are within this range of altitudes. In general, altitude has an influence on the timing of fruit development and on ripening, as was seen in our observed time to harvest. However, prolonged fruit development does not necessarily result in reduced quality [20] (refer to Table 1).

Zone 2, unlike the other two zones, is isothermal; it has a warm climate for 72% of the year, and is sub-humid for most of the year, with an average annual temperature above 22 °C. In all three zones, high temperatures predominate in April and May and precipitation is in the summer. All three zones, especially Zone 1, have a summer period without rain; in Zone 1, it is called Verano de Santiago [21].

The climatic conditions of Zone 2 may be related to the number of seeds; fruit in this zone presented an average of three seeds and a high percentage of viviparity (30%). The fruit of Zone 2 exceed the weights of other studies, but not those of Zone 1 and 3. Atapattu et al. [22] (2014) report a weight of 175 g in Sri Lanka, with an average temperature of 29 °C, and Kong et al. [11] report a weight of 118.09 ± 35.48 g in Malaysia, with an average temperature of 28 °C.

The fruit take longer to reach the semi-ripe stage in Zone 2, where the climate is more stable, and there is less precipitation during the development period. Dussi [20] notes that development time does not affect the quality of fruit, as they appear to accumulate more sugars.

Premature seed germination may be caused by a number of factors. Cota-Sánchez et al. [23] suggest that it may be a result of the local environment and the age of the tree. In addition to these factors, early germination in angiosperms can occur in fruit seeds in the same species but grown in different environments [24].

Temperature is one of the factors that was studied. Penfield and MacGregor [25] also suggest that temperature is a factor in early seed germination. They find that the canistel plant detects high temperature and signals the developing fruit. Canistel fruit from Zone 2 were harvested during the months from May to July, the hottest months (Table 1).

The accumulation of proteins and starch also favors premature germination [26], as demonstrated in broad bean (*Vicia faba*) by Borisjuk et al. [27]. Green canistel has a moisture content of 33.9%, with the pulp measuring 16 °Bx, and 38.9% starch, which could favor premature germination. Farnsworth [28] writes that to understand the physiology of the plant it is necessary to examine the ecology and evolution of the species.

On the contrary, the information gathered from the leaves demonstrates that the first three principal components explain 84.6% of the variability present among the leaves in the three study areas, using only the chosen variables. According to the data, the size, shape, and width/amplitude of leaves are differentiated by their area, perimeter, and minor diameter. Additionally, there is a positive correlation between the area and the minor diameter of a leaf. There are various factors that cause morphological changes in plants (temperature, humidity, altitude, light) [29,30], resulting in variability in shape and size even within the same species. Temperature is one of these factors [31]. In this case, the canistel crops are situated relatively close to each other, but the climatic conditions, particularly temperature, have a direct impact on the development of the plant and its fruiting. Therefore, it is probable that the variations in this species are mainly due to environmental factors.

The cultivation of fruit trees is inherently valuable to the agricultural industry due to its provision of enduring job opportunities, income streams, and potential for other activities or products derived from its produce. Since its inception, the production of fruit species has been of great economic and social importance.

## 5. Conclusions

During the analysis of canistel fruit and leaves in three zones of Mexico, it was discovered that the tree located in Zone 2 (Jiutepec) had the highest probability of propagation and was at the greatest altitude above sea level. This tree had larger fruit size, quantity of pulp and number of seeds, in addition to being sweeter due to a higher amount of °Bx.

The leaves were sturdier, narrower, and lengthier, facilitating our observation of stylized and consistent leaves. By using the three principal variables of leaf area, perimeter, and minor diameter, we can characterize the canistel (*P. campechiana*) leaf to distinguish accessions or cultivars.

This study advances our understanding of this underutilized species, offering insight into selecting the morphological characteristics that enhance its commercial and ornamental value.

**Author Contributions:** Conceptualization, S.E.-L. and K.M.G.-V.; methodology, S.E.-L. and K.M.G.-V.; software, K.M.G.-V. and J.F.P.-B.; validation, S.E.-L., T.R.-G. and J.F.P.-B.; formal analysis, K.M.G.-V.; investigation, S.E.-L., K.M.G.-V., S.L.E.-A. and J.G.C.-C.; resources, S.E.-L.; data curation, writing—original draft preparation, S.E.-L. and K.M.G.-V.; writing—review and editing, S.E.-L., J.F.P.-B., T.R.-G. and K.M.G.-V.; supervision, S.E.-L.; project administration, S.E.-L.; funding acquisition, S.E.-L. All authors have read and agreed to the published version of the manuscript.

**Funding:** This research received no external funding.

**Data Availability Statement:** The data presented in this study are available on request from the corresponding author.

**Acknowledgments:** The authors thank the Secretaría de Investigación y Posgrado (SIP), Program BEIFI of Instituto Politécnico Nacional México, for their support in carrying out the research; and CONAHCyT.

**Conflicts of Interest:** The authors declare no conflict of interest.

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
