# Peer review of "Harvest Season and Morphological Variation of Canistel (Pouteria campechiana) Fruit and Leaves Collected in Different Zones of Mexico"

_horticulturae, doi:10.3390/horticulturae9111214_

Round 1

Reviewer 1 Report

Comments and Suggestions for Authors

Comments for the authors

Horticulturae 2647746: Harvest season and morphological variation of canistel (Pouteria campechiana) fruits and leaves collected in different zones of Mexico.

The subject of this manuscript falls within the general scope of the journal, and the study of the harvest season and morphological variation of canistel (Pouteria campechiana) fruits and leaves collected in different zones of Mexico is relevant.

Keywords: the selected keywords are suitable.

Introduction: this section contains the information that justifies this work. However, I feel that more information about Pouteria campechiana tree and fruits (kind of botanical fruit, actual and potential uses, knowledge about its cultivation, etc), is necessary to further highlights the importance of this study.

Materials and Methods: this section includes details about the methods employed. However, some other details are needed:

Page 2, lines 77-81: This paragraph mixes the description of Table 1 and Figure 1, so it is confused. Clarify each of one. What is the meaning of "zone 1 and zone 2 (2018-2022), each with a tree under backyard cultivation; zone 3 (2022-2023), corresponds to subsidiary 1 of seeds from zone 2"? Clarify the range of years and the concepts of a tree under backyard cultivation (only one tree?) and subsidiary 1 of seeds from zone 2. How old are the trees?

Page 2, Table 1: The legend of Table 1 is incomplete. Each variable must be mentioned. Then, for Temperature there is too much text.

Page 3-4, Figure 1: The legend of Figure 1 is incomplete. Each variable must be mentioned. Which is the title of the Y axis of the left? Then, precipitation must be represented by columns.

Page 4, lines 89-93: The description of this paragraph to which site correspond? Due to Figure 1A and 1B are too similar, an exhaustive description must be made for notice the differences.

Some of the obtained results (early seed germination) were not detailed in Materials and Methods.

Page 4, lines 97-101: "Once the trees to be sampled were located with constant walks and monitoring (fruiting), fruits and leaves were harvested." It is not clear enough if the trees were naturally grown or if they were planted according to the tittle "Planting material..."  This sentence must be clarified. The number of trees sampled per zone were not specified. This information is quite relevant to conclude that the obtained results could be attributed to the ambiental conditions of the zone instead to the genotype, plant age, plant position respect to the light, winds, etc. How were the fruits and leaves harvested? Randomly across the entire tree? Ripe leaves must be replaced by mature leaves.

Page 4, lines 111-113: "Total soluble solids (TSS) which indicate sucrose content and includes carbohydrates, organic acids, proteins, fats, and minerals of the fruit (28 to 62 % Fisher (Japan) refractometer) [12]". This sentence must be corrected as follow "Total soluble solids (TSS) which indicate sucrose content and includes carbohydrates, organic acids, proteins, fats, and minerals of the fruit, were registered using a (28 to 62 % Fisher (Japan) refractometer) [12].

Page 4, line 116: Figure 1 must be replaced by Figure 2.

Results

Some of the obtained results (time from anthesis to fruit harvest as well as harvest season) were not detailed in Materials and Methods.

Page 5, lines 156-158: a brief description of the characteristics of each zone must be included in this section.

Page 6, lines 171 and 173: Figure 2 must be replaced by Figure 3. These variables were not mentioned in Materials and Methods. How were they obtained?

Page 6, Table 3: In this Table, letter a indicates the lowest value, while in Table 2, letter a indicates the highest value. The criterion must be the same for all the Tables.

Discussion

Page 9-10: I feel that this section presents information (data obtained) that must be part of the Results section, as well as the citation of Tables. Also, the discussion of the obtained results is light. I think that this fact is in part due to the methodology employed, i.e. whether the differences found were due to genotype or environmental conditions at the site.

Page 10, Lines 271-273: “in canistel in green state the humidity is 33.9 %, the pulp has 16 °Bx, and starch 38.9 % which could favor premature germination”. Which is the cite for these results?

Page 10, Lines 274-275: Colunga-García-Marín and Zizumbo-Villarreal [26], catalog the canistel as a non-domesticated fruit tree. I think that this cite is more suitable for the Introduction section.

Conclusions

This section is quite similar to the Results and Discussion sections, and must be rewritten, including how this study can contribute to the knowledge of Pouteria campechiana, to its future cultivation, etc.

Final comments: this manuscript needs the corrections suggested before its publication in Horticulturae. Also, a careful revision of the English grammar in needed.

Comments on the Quality of English Language

A careful revision of the English grammar in needed.

Reviewer 2 Report

Comments and Suggestions for Authors

The study provides morphological variations of fruit and leaves of the Pouteria campechiana from three different geographical areas. The study found significant variations among the characteristics of different areas. However, there are some questions and suggestions for the improvement of this study.

The plural of fruit is fruit not the fruits so it should be change throughout the text and title.

Mention in the abstract how many trees were selected and studied in each region.

Which morphometric parameters were studied mention in the abstract.

Line 29-32 the sentence is not clear should be modify.

Line 46-51 the sentence is mixed and beyond the understand of readers please modify it and keep short sentences.

Branch, b small

Total soluble solids (TSS) which indicate sucrose content and includes carbohydrates, organic acids, proteins, fats, and minerals of the fruit (28 to 62 % Fisher (Japan) refractometer). Not clear

Collect a larger number of fruits and leaves from each zone to increase the statistical power and reliability of the results. This will provide a more comprehensive understanding of the morphological variations among different Canistel specimens.

Incorporate molecular techniques, such as DNA sequencing or genotyping, to complement the morphological characterization. Genetic analysis can provide insights into the genetic diversity, relatedness, and population structure of Canistel trees in different regions.

Collect and analyze environmental data, such as temperature, rainfall, soil characteristics, and elevation, for each zone. Or provide details of the environmental data of all regions.

Comments on the Quality of English Language

Mentioned in comments to authors

Round 2

Reviewer 2 Report

Comments and Suggestions for Authors

The study is well revised but still there are still some deficiencies which are highlighted in the following comments

In many places still fruits are not changes to fruit.  

Check line 44-45 species names are not italic. All species names must be italic. Check the whole MS.

Check typo in line 16

Check line 77-78

Line 41-43 should be cited with recent study. https://doi.org/10.1016/j.chnaes.2021.08.002

Line 283 “There are various factors that cause” specify the factors with related studies.

Comments on the Quality of English Language

Check typo in line 16

Check line 77-78
